# A research on cross-age facial recognition technology based on AT-GAN

**Guangxuan Chen**[◉], **Xingyuan Peng**[iD][◉]*, **Ruoyi Xu**[◉]

School of Information Network, Zhejiang Police College, Hangzhou, Zhejiang Province, China

◉ These authors contributed equally to this work.
* pengxingyuan040319@163.com

## Abstract

Currently, predicting a person's facial appearance many years later based on early facial features remains a core technical challenge. In this paper, we propose a cross-age face prediction framework based on Generative Adversarial Networks (GANs). This framework extracts key features from early photos of the target individual and predicts their facial appearance at different ages in the future. Within our framework, we designed a GAN-based image restoration algorithm to enhance image deblurring capabilities and improve the generation of fine details, thereby increasing image resolution. Additionally, we introduced a semi-supervised learning algorithm called Multi-scale Feature Aggregation Scratch Repair (Semi-MSFA), which leverages both synthetic datasets and real historical photos to better adapt to the task of restoring old photographs. Furthermore, we developed a generative adversarial network incorporating a self-attention mechanism to predict age-progressed face images, ensuring the generated images maintain relatively stable personal characteristics across different ages. To validate the robustness and accuracy of our proposed framework, we conducted qualitative and quantitative analyses on open-source portrait databases and volunteer-provided data. Experimental results demonstrate that our framework achieves high prediction accuracy and strong generalization capabilities.

## 1. Introduction

The appearance of a person's face changes over time, with features such as the Mongolian fold rate decreasing, the slant of the eye opening becoming more horizontal, and eye color becoming lighter, leading to significant changes in appearance and making face matching for identification purposes very difficult. Through deep learning algorithms, it is possible to train models to recognize and predict the changing facial features of individuals over time, even accurately identifying specific individuals from large datasets years later. However, the application of this technology is not without challenges. The facial features of children change significantly as they grow, requiring the model to be able to handle fine-grained age-related facial changes. Additionally,

**Data availability statement:** All relevant data are within the manuscript and its Supporting Information files.

**Funding:** The author(s) received no specific funding for this work.

**Competing interests:** The authors have declared that no competing interests exist.

the problem of extracting effective information from old, low-resolution photos due to historical photo quality and availability issues must also be overcome.

In this paper, we present a novel cross-age face recognition framework by integrating face aging transformation features, deep learning-based face recognition techniques, and old photo restoration technologies. This innovative approach is specifically designed to address the critical societal challenge of improving the success rate of locating missing children through robust and accurate face matching across different age stages. The core contribution of this work lies in the development of a comprehensive and multi-faceted methodology that synergistically combines three key components: (1) the Attention and Transition Generative Adversarial Network (AT-GAN), a face prediction model that captures subtle facial aging patterns with high precision; (2) advanced old photo restoration techniques, which enhance the quality and usability of historical or low-resolution images; and (3) face recognition algorithms, optimized for cross-age scenarios. Together, these components form a unified framework capable of addressing the inherent challenges of cross-age face recognition, such as large temporal gaps, facial feature variations, and image quality degradation.

The paper provides a detailed theoretical and technical exposition of the proposed framework, including the architectural design of the AT-GAN model, the integration of photo restoration modules, and the optimization strategies for face recognition. Furthermore, we develop a fully functional system to implement the AT-GAN-based face prediction algorithm, demonstrating its practical applicability and scalability. To validate the effectiveness of our approach, we conduct extensive experimental analyses using diverse datasets. The results are compared against several baseline methods and existing state-of-the-art techniques, highlighting the superior performance of our framework in terms of accuracy, robustness, and generalization capability. Notably, our method achieves significant improvements in cross-age face matching accuracy, particularly in scenarios involving long-term age progression and low-quality input images.

## 2. Related work

Currently, in the field of cross-age recognition, experts and scholars at home and abroad have conducted a series of research, mainly focusing on the following aspects.

### 2.1 Cross-age face recognition technology

Cross-age face recognition technology is an important research direction in the field of face recognition, aiming to solve the recognition problems caused by changes in facial features with age. In recent years, cross-age facial recognition technology has made significant progress. Traditional cross-age face recognition methods mainly rely on manually designed feature extraction and matching strategies. For example, a method using a combination of Gradient Oriented Pyramid (GOP) and Support Vector Machine (SVM) has achieved good performance in the passport photo verification task [1]. However, these methods often have difficulty handling cross-age recognition

problems in complex environments because they cannot effectively capture the deep features of faces that change over time. With the rise of deep learning technology, cross-age face recognition methods based on deep learning have begun to receive widespread attention. These methods automatically learn and extract facial features that are invariant across ages by building deep neural networks. For example, scholars such as Jian Zhao proposed a deep age-invariant model (AIM), which achieves significant performance improvements by jointly learning cross-age face synthesis and recognition tasks [2]. In addition, there are also studies that introduce disentangled representation learning and propose a disentangled representation learning network based on generative adversarial networks to achieve more accurate cross-age face recognition. In addition, some research also attempts to solve the cross-age face recognition problem by synthesizing face images of different age groups. For example, scholars such as CHENG Zhikang proposed a cross-age face synthesis method based on conditional adversarial autoencoders [3]. This method can effectively avoid the deformation and distortion of facial organs and maintain the stability of local structural features of the face. In addition, how to effectively fuse hard biometric and soft biometric information, and how to further improve the generalization ability and real-time performance of the model, are still key issues that need to be solved in future research. Cross-age face recognition technology is a complex but extremely challenging research field. In the realm of multi-task learning, a model proposed in 2024 jointly optimizes age prediction and identity classification tasks, enhancing the model's generalization capability through feature fusion (e.g., combining LBP texture features with deep learning global features) [4]. In terms of feature disentanglement techniques, researchers in 2023 proposed decomposing facial features into pure age features, pure identity features, and shared features, achieving 97.53% accuracy on the Age-DB30 dataset by employing a two-stage constraint algorithm to enhance feature independence. Similarly, a model introduced by scholars incorporates an attention mechanism, effectively preventing the loss of identity information when removing age-related features. In the area of hybrid architectures and novel network designs, scholars have proposed combining CNNs with Transformers. The CvT model integrates convolutional neural networks with Transformers, preserving more image details and addressing the issue of information loss caused by feature compression in traditional models. Additionally, cross-age encoding methods, such as Cross-Age Reference Coding (CARC), have been proposed, which utilize images of the same individual at different ages to construct a reference feature space, generating age-invariant encodings that significantly improve retrieval accuracy. Through continuous exploration and innovation, combined with deep learning and other advanced technologies, more accurate and robust cross-age face recognition solutions are emerging. In addition, Generative Adversarial Networks (GANs) have emerged as a powerful tool for cross-age face recognition due to their ability to generate realistic facial images and model age-related variations.

## 2.2 Generative adversarial network

GANs represent a deep learning framework composed of two key components: the Generator and the Discriminator. The Generator takes random noise as input and produces synthetic samples that mimic the distribution of real data, while the Discriminator is tasked with distinguishing between these generated samples and authentic ones. Through an adversarial training process, the two engage in a dynamic competition: the Generator aims to "deceive" the Discriminator, while the Discriminator continuously refines its ability to differentiate. This adversarial interplay ultimately enables the Generator to produce highly realistic data, such as images, audio, or text. The core training mechanism of GANs is based on a minimax game. The Generator's objective is to minimize the Discriminator's accuracy in identifying fake samples, whereas the Discriminator strives to maximize its ability to distinguish between real and fake samples. By alternately optimizing their respective loss functions (such as the cross-entropy loss in the original GAN or the Earth Mover's Distance in Wasserstein GAN), the model gradually converges to an equilibrium state where the distribution of generated samples closely aligns with that of real data. However, challenges such as mode collapse (where generated samples become overly homogeneous) and unstable gradients often arise during training, necessitating techniques like weight clipping and gradient penalty for optimization. To address diverse application scenarios, various enhanced versions of GANs have been developed.

For instance, Conditional GANs (cGANs) incorporate label information to guide content generation, CycleGANs facilitate cross-domain translation without paired data, and StyleGANs allow for detailed control over generated outputs through style vectors. These variants find extensive use in fields like image synthesis, data augmentation, and medical imaging. Nonetheless, challenges related to training complexity, computational resource demands, and controllability of generated outputs remain critical areas for future research.

Since GANs were proposed in 2014, they have shown great potential and application value in many fields. Not only can they generate high-resolution images, they can also play an important role in semi-supervised learning [5], conditional generation, and private data enhancement. In addition, variants of GANs, such as Wasserstein GAN (WGAN), Deep Convolutional Generative Adversarial Network (DCGAN), Softmax GAN, etc., further expand the application scope and performance of GANs. Although GANs have made significant progress in many aspects, they still face some challenges during the training process, such as training instability and mode collapse. To address these issues, researchers have proposed various improvement strategies. These include optimizing the gradient update process with the Adam algorithm, leveraging geometric structures to better analyze the GAN training process [6], and employing different loss functions and training techniques to enhance the model's stability, performance, and generation quality [5].

GANs have been effectively utilized to tackle cross-age recognition challenges through age progression and regression, identity-preserving age synthesis, disentangled representation learning, and data augmentation. For instance, GAN-based frameworks like IPGAN enforce identity consistency during age synthesis, while others disentangle age and identity features into separate latent spaces. Additionally, GANs generate diverse age-specific facial images for data augmentation and domain adaptation, bridging domain gaps and improving recognition accuracy across age groups.

## 2.3 Old photo restoration technology

Old photos may suffer from various damages due to the passage of time, poor storage conditions, or other reasons. Old photo restoration technology is a technology designed to restore and improve the quality of old photos. With the development of computer vision and deep learning technology, old photo restoration technology has made significant progress. Old photos of missing children can be restored to make subsequent facial simulations more robust. Traditional old photo restoration methods mainly rely on image processing technology, such as image inpainting technology based on geometric image models and image completion technology based on texture synthesis [7]. These methods have their own advantages in dealing with small-scale defects and large-scale missing blocks, but they usually require local or global modifications to the image, which may lead to the loss of the naturalness and authenticity of the image. In recent years, deep learning-based methods have shown great potential in the field of old photo restoration. For example, a method based on Generative Adversarial Networks (GAN) was proposed for the repair of damaged old photos. This method improves the repair effect through local convolution operations and contextual attention modules, and can handle any non-center irregular damaged areas.. In addition, some research has proposed an old photo restoration method using deep learning and image sharpening strategies. This method can improve the clarity of old photos and repair damaged areas according to the proportion of portraits. In addition to the above methods, some studies have proposed comprehensive repair frameworks that combine multiple technologies. For example, the Pik-Fix framework can address various degradation problems in old photos, as well as perform color correction and coloring. This approach significantly improves both repair efficiency and final image quality[8]. In addition, the Stacked Median Restoration plus (SMR+) method demonstrates the effectiveness of deep learning in dealing with complex degradation problems by utilizing the symmetry of stereo photos to repair historical stereo photos.

## 2.4 Attention mechanism

The attention mechanism is a technology that simulates the human visual selectivity mechanism. It aims to filter out information that is more relevant and critical to the current task goal from a large amount of information, so as to improve the efficiency and accuracy of information processing. It has been widely used in the field of deep learning, especially in

                                                                

natural language processing, image recognition, recommendation systems and many other aspects. In terms of visual attention, studies have shown that the human visual system has limited information processing capabilities and can filter out irrelevant information by selectively focusing on certain stimuli [9]. This mechanism not only helps to improve resource processing efficiency, but also enhances the model's attention to important features, thereby improving the performance of the model [10]. For example, the introduction of the attention mechanism in the Convolutional Neural Network (CNN) can improve the classification accuracy by focusing on the key areas in the image. In the field of Natural Language Processing (NLP), the attention mechanism also plays an important role. It allows the model to dynamically adjust the degree of attention to different words or phrases when generating text or classifying text, so as to better understand the meaning of the text. In addition, the attention mechanism is also used to improve the performance of the recommendation system by assigning different weights to different user features, so that the recommendation system can more accurately predict user preferences. The effect of the attention mechanism does not only come from its intuitive visual attention explanation, but is closely related to the way it is implemented in the model. For example, the application of feature map multiplication in the attention mechanism can significantly improve the learning and generalization capabilities of convolutional neural networks [10]. In addition, the attention mechanism can be applied to the input and output of the model in different ways to further improve the performance of the model. In the field of image recognition, the attention mechanism significantly improves the performance of CNNs by simulating the ability of the human visual system to focus on important information. Specific case studies have shown that the attention mechanism can effectively improve the model's attention to key features, thereby improving the accuracy and efficiency of image recognition.

### 2.5 Face aging dataset

The research and application of face aging datasets has become an important branch in the field of computer vision, involving age estimation, cross-age face recognition, face aging/de-aging, and many other aspects. Since its release in 2004, the FG-NET aging database has been widely used to support research activities aimed at understanding the changes in facial appearance caused by aging [11]. This database not only promotes the development of research on age estimation, age-invariant face recognition, and age progression, but also provides guidance for future research trends, requirements, and research directions. In addition, research on cross-age face recognition has also shown that facial features change with age, which affects recognition accuracy [12].

## 3. Methods

Ethics Statement: All participants have agreed to have their images published together with the manuscript, as these images will not disclose any other information about the participants.

The individual in this manuscript has given written informed consent (as outlined in PLOS consent form) to publish these case details.

### 3.1 Research route

The goal of this work is to restore the predicted appearance of a person years later or throughout their lifespan based on an old photograph of a specific person at a specific age, and then use face recognition technology to find the target person in a specific database. The entire process includes steps such as image blurring, crack repair, intelligent coloring, facial prediction, and face matching.

The proposed solution is roughly divided into three steps, as shown in Fig 1.

The first step is to restore the old photos. These old photos may be damaged due to the passage of time, poor preservation conditions or other reasons, or some old photos are black and white. This paper proposes an old photo restoration technology based on machine learning to perform intelligent restoration and intelligent coloring of the original photos, which can restore the original photos to pictures and data that are more suitable for face prediction.

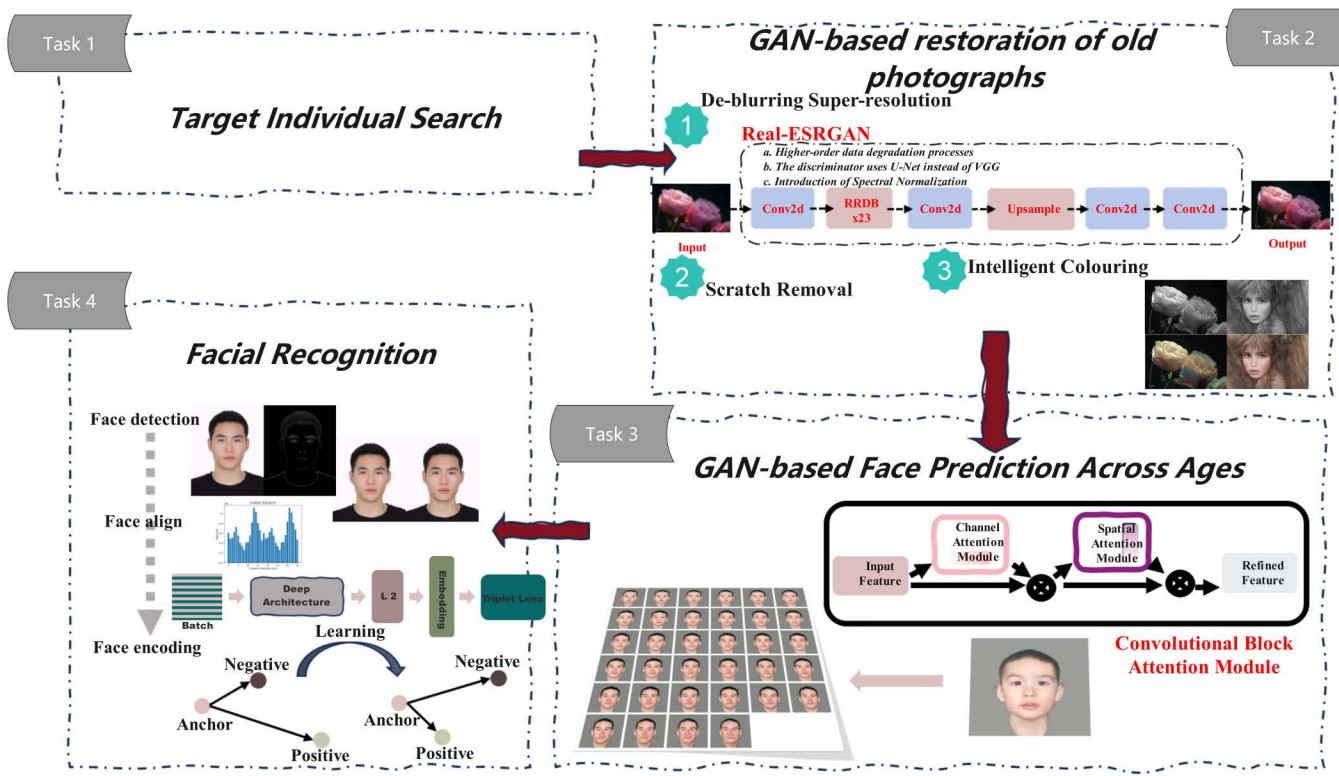

**Fig 1. Technology Roadmap.**

The second step is to perform face simulation prediction on the restored photos. Based on the GAN generative adversarial network for machine learning, a new generative adversarial network architecture proposed by scholars such as Roy Or-El and Soumyadip Sengupt[13] consists of a single conditional generator and a single discriminator. The conditional generator is responsible for the conversion across age groups and consists of three parts: identity encoder, mapping network and decoder. However, the training range of the model dataset is small, and the faces are mostly white, which is not suitable for the features of yellow faces. To this end, this paper uses more ethnic datasets for further training. On this basis, the self-attention mechanism is improved and the AT-GAN model is proposed to improve the success rate of face prediction. The self-attention mechanism can find key features of key parts such as eyes and noses that do not change much to screen the results of multiple iterations, thereby enhancing robustness.

The third step is to find the specific correct person in the face library through face recognition, which involves face detection, face alignment, face encoding, and identity recognition. By finding the person in the face library, it simulates the actual use of the project and can be used to evaluate the generalization ability of the model.

### 3.2 Old photo restoration

**(1) Photo deblurring.** As old photos are very old, some of them will be severely degraded, and the pixels will be far lower than those presented by current advanced camera technology. Therefore, this paper decides to use GAN to deblur the old photos and improve their resolution as much as possible. The study in the literature [14] proposed a joint framework that combines the Generative Facial Prior Generative Adversarial Network (GFP-GAN) and Real-Enhanced Super-Resolution Generative Adversarial Network (Real-ESRGAN) methods, which provides a good idea, as shown in Fig 2. It is more effective than traditional bicubic interpolation in simulating real-world image degradation by implementing

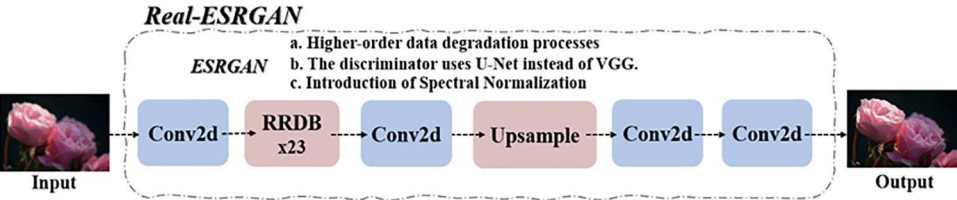

**Fig 2. Real-ESRGAN.**

a high-order degradation model (HDM) [15]. However, the images reconstructed by Real-ESRGAN have the problems of over-smoothing and loss of texture information, which makes its performance inferior to classic models such as Super-Resolution Generative Adversarial Network (SRGAN) and Enhanced Super-Resolution Generative Adversarial Network (ESRGAN) [16]. In order to solve these problems, an improved image super-resolution model IRE is proposed. This model enhances the texture details of the reconstructed image by removing the first-order degradation model of HDM and introducing a channel attention mechanism and a SmoothL1 loss function [16].

In addition, research on GAN-based image restoration algorithms shows that GAN's superiority in image detail generation has made it applicable to image deblurring and image super-resolution problems. In particular, the quality of image restoration can be further improved by introducing an attention mechanism and optimizing the loss function. These research results support the combined use of GFP-GAN and Real-ESRGAN, as they both involve using deep learning techniques to improve image quality, especially when dealing with severely degraded images.

This paper achieves superpixel processing of old photos through the enhanced training model of GAN.

**(2) Image crack removal.** The datasets about image scratches mainly include NVIDIA Irregular Mask Dataset (NVIDIA-IMD), II-CGAN and other datasets. This paper will refer to the supervised learning model method of the Multi-Scale Feature Attention (MSFA) network, and propose a Semi-MSFA scratch repair method that comprehensively utilizes synthetic datasets and semi-supervised learning of real old photos to better suit the task of old photo restoration. Semi-MSFA involves the study of algebraic structures and can also be applied to the field of image processing, especially in the study of multispectral image demosaicing algorithms. Semi-MSFA is not only used in theoretical research, but also has its value in solving practical problems, and can also be used in the study of image scratch removal.

The idea of model iteration and supervised analysis of photos are very important in image processing. They can supervise the model to make the image as close to the true value as possible after passing through the network. This paper adopts three loss functions for supervision.

This paper adopts three loss functions in its supervision method.

Loss of Mean Square Error, $L_{\{mes\}}$ is.

$$L_{\{mse\}} = \frac{1}{I} \sum_{i=1}^{I} \left\| I_{gt}^i - M_{MSFA}\left(I_{syn}^i\right) \right\|_2 \tag{1}$$

Perceived losses $L_{per}$ is.

$$L_{per} = \frac{1}{I} \sum_{i=1}^{I} \left\| \varnothing\left(I_{gt}^i\right) - \varnothing\left(M_{MSFA}\left(I_{syn}^i\right)\right) \right\|_2 \tag{2}$$

Confrontational loss $L_{gan}$ is.

$$L_{gan} = E\left[\log_a D\left(I_{gt}^i\right)\right] + E\left[\log_a D\left(1 - I_{syn}^i\right)\right] \tag{3}$$

**(3) Intelligent colouring.** After superpixel processing and scratch removal, this paper will perform intelligent colorization of the image. Color images are more conducive to facial prediction, as shown in Fig 3.

With the application of deep learning technology, deep learning-based image colorization methods have begun to emerge. These methods use the powerful learning ability of CNNs to predict and propagate color information by training a large number of image data sets, thereby achieving automatic colorization. In particular, the application of advanced technologies such as GANs and Pix2Pix models makes the image colorization effect more realistic and natural [16].

This paper uses the GANs model to test the experimental images, and it can be found that the colorization effect of the portraits and objects in the images is good, which can be applied to the subsequent operations of facial prediction.

### 3.3 GAN-based face recognition across ages

**(1) Generative adversarial networks.** As introduced earlier, GANs are a deep learning model based on adversarial learning between a generator and a discriminator. During the training process, the generator network *G* aims to generate realistic images to deceive the discriminator network *D*, while *D* strives to distinguish between real images and those generated by *G*. This adversarial process, inspired by the two-person zero-sum game in game theory, leads to the continuous optimization of both networks, enabling *G* to eventually produce high-quality data samples [17]. The objective function of GAN is as follows:

$$\min_G \max_D V(D, G) = E_{x \sim p_{data}} \left[ \log D(x) \right] + E_{z \sim p_z(z)} \left[ \log \left( 1 - D \left( G(z) \right) \right) \right] \tag{4}$$

The proposed GAN model in this manuscript is illustrated in Fig 4.

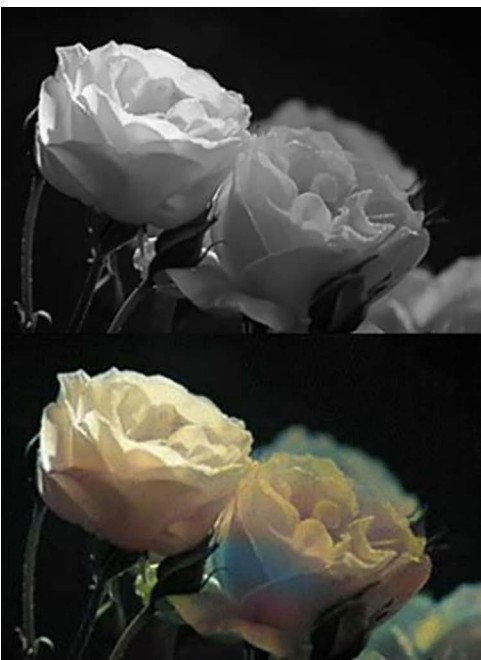

**Fig 3. Image Colouring Comparison Chart.**

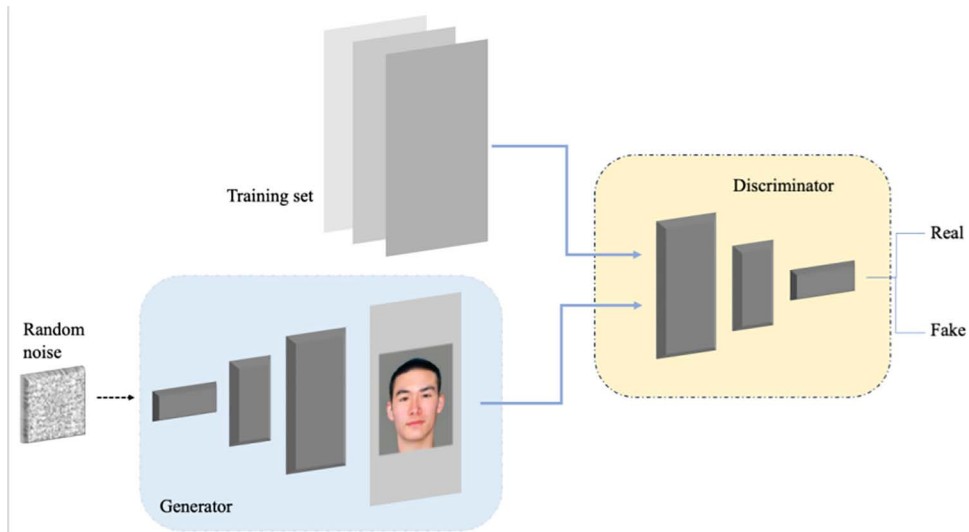

**Fig 4. GAN model.**

**(2) Database introduction.** This paper further processes the open source dataset FFHQ-Aging and annotates it with age, gender, and semantic segmentation. Fixed age categories are used as anchors to approximate continuous age transformations. The screening criteria are gender confidence less than 0.66, age confidence less than 0.6, head yaw angle greater than 40, head spacing angle greater than 30◦, black glasses label and eye occlusion, and eye pair scores greater than 90 and 50. In summary, there are 28,692 males and 33,018 females in the dataset, including age label information for portraits.

**(3) Model network design.** Regarding the design of the model network, this paper introduces the self-attention mechanism. The attention mechanism can help the model focus on the key parts of the input image, thereby improving the accuracy of identity preservation. In the age transformation task, this means that the model can better identify and retain the unique features of an individual, such as eyes, nose, mouth, etc., which are relatively stable throughout a person's life cycle. At the same time, a generative adversarial network is introduced for face prediction, named AT-GAN, which aims to improve the accuracy of face prediction. After multiple sets of data are obtained through GAN iterations, they are screened through the self-attention mechanism to obtain more accurate face prediction results.

The design architecture of the generative adversarial network is as follows:

Identity encoder $E_{id}$ accept input image x and extract identity feature tensor $w_{id} = E_{id}(x)$. These features contain information about the local structure of the image and the general shape of the face, which play a key role in generating the same identity. The identity encoder contains two downsampling layers followed by four residual blocks.

Mapping network M, $Z \rightarrow W_{age}$ embedding age input vectors into a uniform age potential space $W_{age}$, $W_{age} = M(z)$, where M is an 8-layer network, $W_{age}$ is a potential vector of 256 elements. The mapping network learns an optimal age potential space that enables smooth transitions and interpolation between age clusters, which is necessary for continuous age transformations.

For identity coding, four stylized convolutional layers are used for identity coding, and two up-sampled stylized convolutional layers are used to generate the original size image. The overall generator mapping from the input image $z_t$ and target age vector $z_t$ to the output image $y$ is as follows.

$$y = G(x, z_t) = F(E_{id}(x), M(z_t)) \tag{5}$$

The age encoder forces the input image x to be mapped to its correct location in the age vector space Z. It produces an age vector corresponding to the source age clusters of the image $x, z_s = E_{age}(x)$. The age encoder needs to capture more global data to encode general appearance regardless of identity.

The StyleGAN discriminator, based on a small batch of standard deviations, modifies the last fully connected layer to have several outputs to distinguish between multiple classes as suggested by Liu et al. For real images from the first category, only the first output is penalized. Similarly, for generated images from the first category, only the first output is penalized.

The introduction of the self-attention mechanism brings three advantages in model improvement. First, the attention mechanism can enhance the accuracy of feature extraction. For example, in the recognition of gender attributes around the eyes based on the attention mechanism, the accuracy of the recognition is improved by strengthening the extraction of feature information around the eyes. This shows that the attention mechanism can help the model pay more attention to the key information in the image, thereby improving the accuracy of recognition. The second is that the attention mechanism enhances the model's ability to identify individual attributes by constructing an Attribute Attention Map (AAM). For example, AANet significantly improves the performance of person re-identification by integrating key attribute information into a unified learning framework [18]. This shows that the attention mechanism can effectively utilize individual attribute information to enhance the identification ability of the model. Third, the attention mechanism can also solve the problem of cross-view consistency and improve the model's ability to distinguish highly similar individuals. Attention-Aligned Network effectively reduces the impact of background clutter and improves feature learning capabilities by introducing the Omnibearing Foreground-aware Attention (OFA) module and Attention Alignment Mechanism (AAM). This further proves the effectiveness of the attention mechanism in improving model generalization capabilities and handling complex scenarios.

The CBAM model used in this article is a model that combines channel attention and spatial attention, aiming to enhance the convolutional neural network's ability to focus on images, as shown in Fig 5.

In the process of facial reconstruction, this paper analyzes the loss function of the model. During the training process, the loss function will be minimized, including:

Adversarial loss: This is the most common and necessary loss in GAN model training and is called the adversarial loss. shows the value obtained by feeding the image into the discriminator, shows feeding a noise into the value returned by the generator.

$$L_{GAN} = E_{x \sim P_{data(x)}} [\log D(x)] + E_{z \sim P_{noise(z)}} [log(1 - D(G(z)))] \tag{6}$$

**Fig 5. Model CBAM.**

Self reconstruction loss: Self reconstruction loss is used to measure the difference at the time of transformation, which is minimised in this paper when the given target age clusters are the same as the source clusters, and is performed with the help of distance to measure the difference between the original and reconstructed data.

$$L_{rec}(G) = \|x - y_{rec}\|_1 \tag{7}$$

Cycle loss of consistency (Cycle loss): To help maintain identity and consistent skin colour, cycle loss of consistency is used in this paper.

$$L_{cyc}(G) = \|x - y_{cyc}\|_1 \tag{8}$$

Identity feature loss: In order to ensure that the generator maintains the identity of the person throughout the aging process, this paper minimises the distance between the identity feature of the original image and the identity feature of the generated image.

$$L_{id}(G) = \|E_{id}(id) - E_{id}(y_{gen})\|_1 \tag{9}$$

Age vector loss: Forces the real and generated images to be correctly embedded into the input source and target age space by penalising the distance between the age encoder output and the age vector $z_s, z_t$. The loss is defined as.

$$L_{age}(G) = \|E_{age}(x) - z_s\|_1 + \|E_{age}(y_{gen}) - z_t\|_1 \tag{10}$$

The overall optimisation function for the face reconstruction process is as follows:

$$\min_G \max_D L_{adv}(G, D) + \lambda_{rec}L_{rec}(G) + \lambda_{cyc}L_{cyc}(G) + \lambda_{id}L_{id}(G) + \lambda_{age}L_{age}(G) \tag{11}$$

**(4) Facial recognition matching.** In the previous stages, we obtained the images generated by face prediction. Next, we find the images of specific people from the database through face recognition comparison. The specific steps are as follows:

Face Detection. First, we need to find the location of all faces in the image or video frame and cut out the image of the face part. We can use the Histogram of Directed Gradients (HOG) to detect the location of the face. First, grayscale the image because color has no obvious effect on finding the location of the face. Then calculate the gradient of each pixel in the image. By transforming the image into HOG form, the project research can extract the features of the image and obtain the location of the face, as shown in Fig 6. The left is the image of the open source website, and the right is the result after the grayscale program.

Face Alignment: The face in a picture may be tilted or just in profile. In order to facilitate face encoding, the face needs to be aligned into the same standard shape. The first step of face alignment is to estimate the feature points of the face. With the help of Dlib's special functions and models, this article can locate the 68 feature points of the face, as shown in Fig 7. The left is an open source website image, and the right is the feature point extraction result. After finding the feature points, the geometric transformation of the image (affine, rotation, scaling) can be used to align the feature points (move the eyes, mouth and other parts to the same position). In this way, the face information in the face library or real-time input can be processed and better compared with the results of facial prediction.

Face Coding: In this study, the FaceNet deep learning model is employed to achieve efficient face recognition and clustering by mapping face images into a compact Euclidean space, where the distance between points directly reflects the degree of face similarity. Specifically, the model generates a 128-dimensional feature vector for each input face image

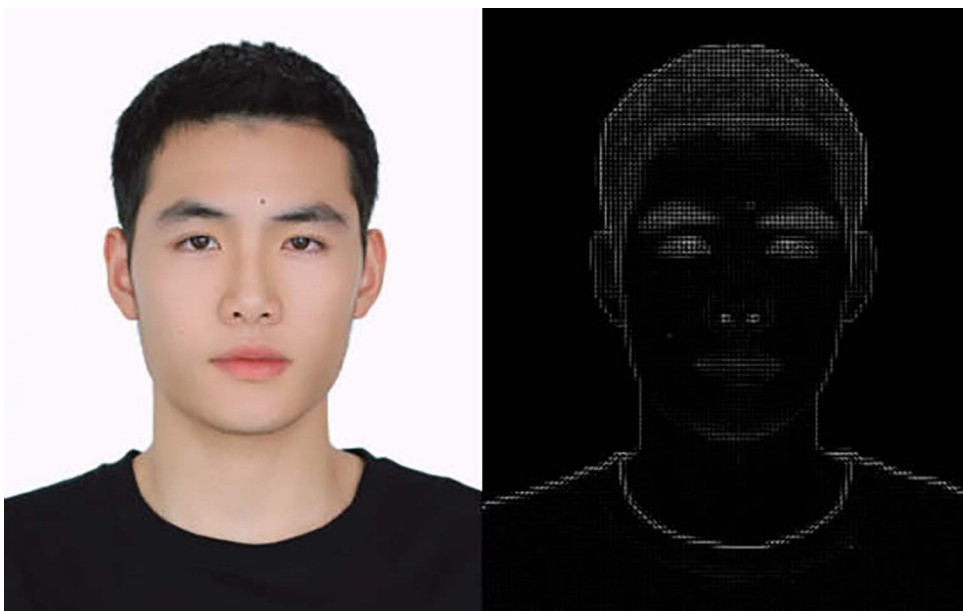

**Fig 6. HOG grey scale map.**

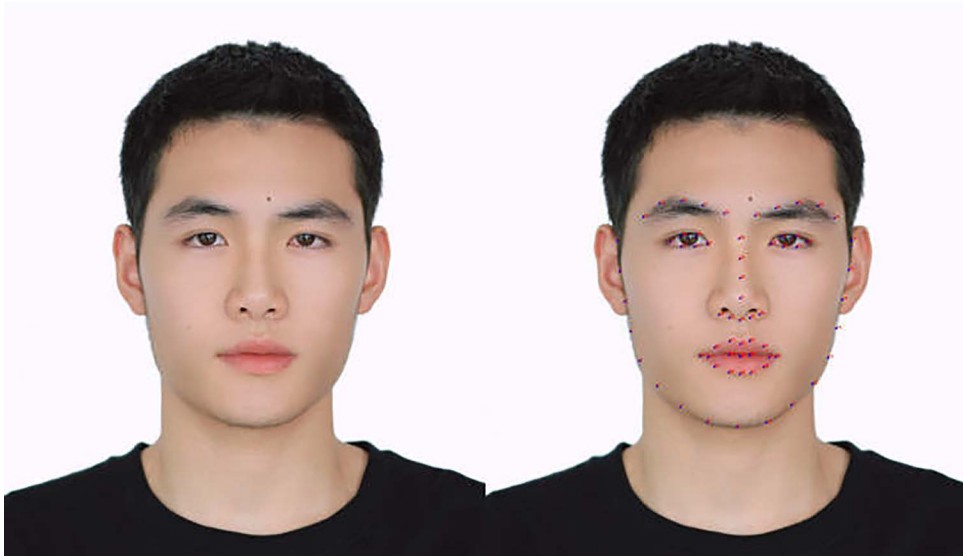

**Fig 7. Face feature point extraction.**

and utilizes the triplet loss function to measure the distance error between samples during training, as illustrated in Fig 8. The training process optimizes the model using stochastic gradient descent, aiming to minimize the intra-class distances while maximizing the inter-class distances, thereby converging to an optimal solution. Through this embedding learning approach, the feature extraction network's output layer is further refined, enhancing the discriminative power of the extracted features. The algorithmic framework of the FaceNet model is depicted in Fig 9.

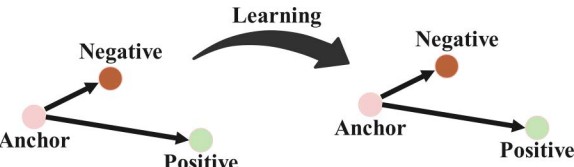

**Fig 8. After learning, positive samples are closer together and negative samples are further apart.**

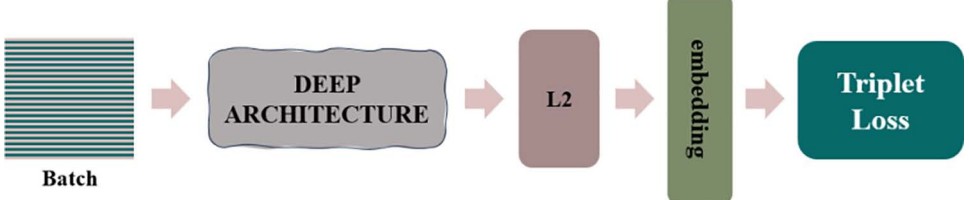

**Fig 9. FaceNet System Framework.**

Face Database Preparation and Recognition Process: The faces of the target population are preprocessed and stored in a face database. Each face image is encoded into a 128-dimensional feature vector using the aforementioned neural network and subsequently saved. During the recognition phase, an input face is transformed into a 128-dimensional vector and compared against the entries in the face database. The comparison is based on the Euclidean distance metric, where the face with the smallest Euclidean distance within a predefined threshold is identified as the match. This process minimizes the value of the Euclidean distance objective function, as mathematically formulated below.

$$d_{Face} = \sqrt{\sum_{i=1}^{n}(x_i - y_i)^2}$$

(12)

## 4. Results

The method of cross-age face recognition technology based on AT-GAN proposed in this paper is introduced in detail in the above sections. The following part will test the method through a series of quantitative and qualitative experiments. The facial photos used in this study were sourced partly from public facial recognition databases and partly from volunteers and the authors of the paper. All experiments conducted were carried out with the consent of all volunteers and the authors, ensuring no ethical or financial conflicts of interest were involved.

In the experiment, firstly, an old photo is restored, then the facial changes are simulated through the AT-GAN model of this paper, followed by face recognition of the generated results, and finally match them against a facial database. The experimental result diagram is shown in Fig 10.

### 4.1 Quantitative model evaluation

To quantitatively evaluate the prediction and recognition performance of the model, this study will use the following method: select 50 volunteers, use their clear childhood photos to predict their facial images at the current age, and perform face matching in a dataset containing recent photos of these volunteers. The dataset includes 500 photos. This process is repeated 5 times to ensure the reliability of the results.

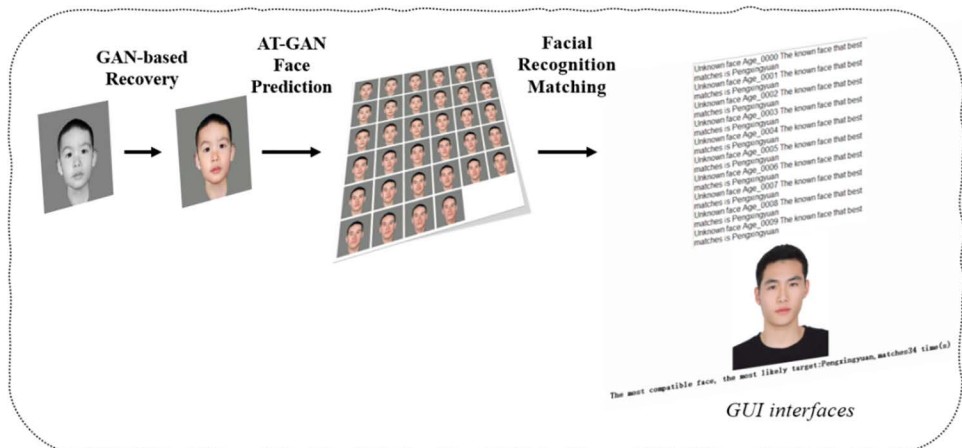

**Fig 10. Experimental results.**

In each matching process, this paper calculates the cosine similarity between the query image (i.e., the predicted facial image at the current age) and all reference images in the dataset. Then, sort them from high to low according to the similarity score, and the higher the ranking, the greater the similarity.

As shown in Fig 11 below, the horizontal axis is the ranking of the original image in descending order, and the vertical axis is each volunteer. In the 5 tests, most of the volunteers' data will be concentrated in the first and second place, proving that the prediction and recognition ability of this model has high accuracy.

## 4.2 Performance analysis

To test the performance of the model, this paper selects the commonly used face aging prediction algorithms currently on the market for comparative analysis.

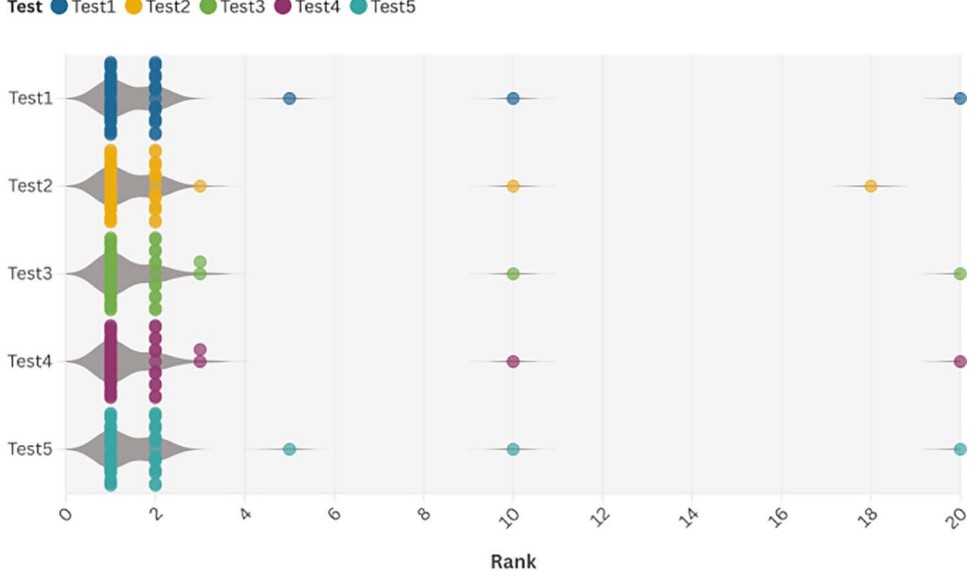

**Fig 11. Test results.**

Of course, this evaluation also has certain limitations. Due to the influence of photo clarity, photo tilt angle, and data set size in actual combat, the accuracy cannot reach the results of experimental tests.

### 4.3 Qualitative model evaluation

In order to test the performance of the model, this paper selects the commonly used face aging prediction algorithms on the market for quantitative comparative analysis.

The T*T* model is a common elderly appearance prediction model, but it is based on traditional algorithms and only provides a simple simulation of facial wrinkles. The age span is single and cannot truly reflect the overall changes of the face at different ages. It has weak generalization ability, weak robustness, and general effect, and is difficult to be effectively used in actual scenarios, as shown in Fig 12.

AgingGer is another popular face-aging commercial app on the Apple Store, which has three effect modes as shown in Fig 13. However, in the tests conducted for the project study, some problems were found with the aging photos generated through AgingGer. Specifically, the application presented relatively severe facial changes and the effects were overly exaggerated and detached from the reality of real facial changes. Compared to reality, the aging effects generated in

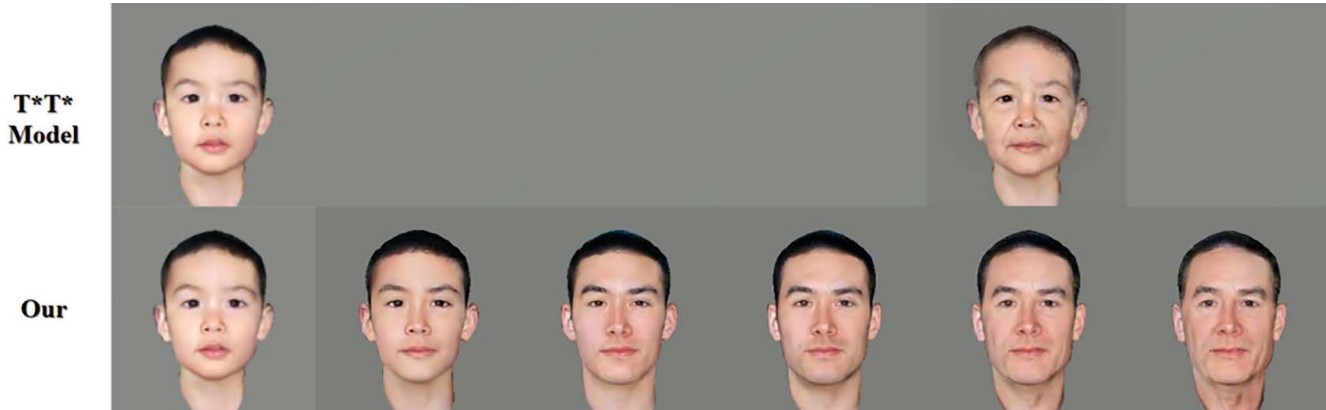

**Fig 12. Comparative analysis with the T*T* model.**

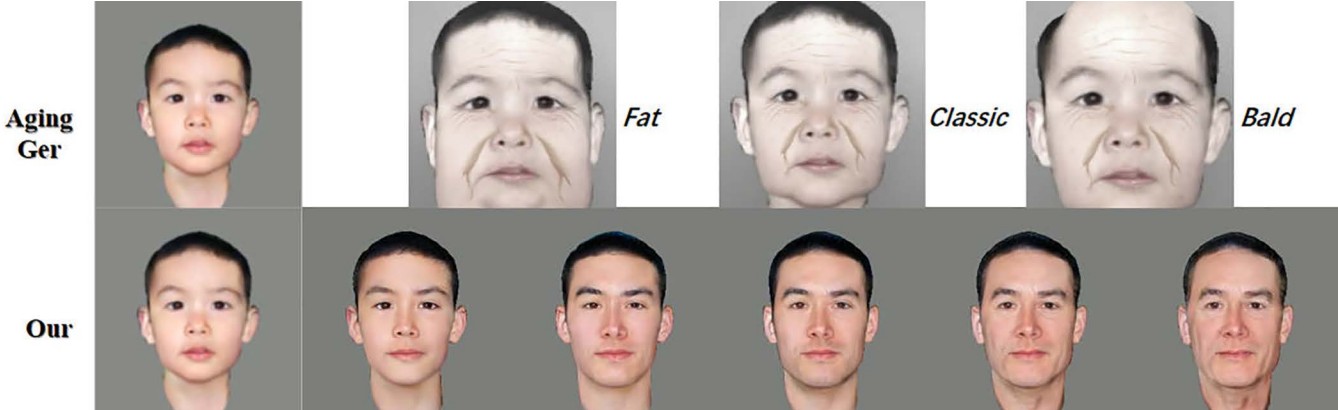

**Fig 13. Comparison with AgingGer.**

AgingGer appear to be exaggerated, and cannot provide a more realistic and accurate prediction of facial looks, making it difficult to play a greater role in practical applications.

Kualian is a photo manipulation app that brings together a wide range of trendy style effects, including a time machine feature. In the test, photos were generated using Kualian for ages 5, 45, and 85 respectively as shown in Fig 14. It can be observed that the models generated for each age group are the same, with some unnecessary items added, which has an impact on the predictive analyses. It should be emphasized that Kualian mainly focuses on image effects and trendy styles, which are more inclined to provide interesting visual effects rather than real face change predictions. In practical applications, these additional effects and items may interfere with the accurate prediction of facial age.

**4.4 Old Photo restoration optimisation**

Old photo processing has an important role in face prediction at a later stage, this paper compares the control variables of old photo processing. One group of data directly through black and white photos for face prediction, and another group technology first through old photo restoration technology, crack repair, intelligent coloring operation, and then face prediction, the two groups of data to compare the accuracy of the results. As shown in Fig 15 below, the lack of the old photo

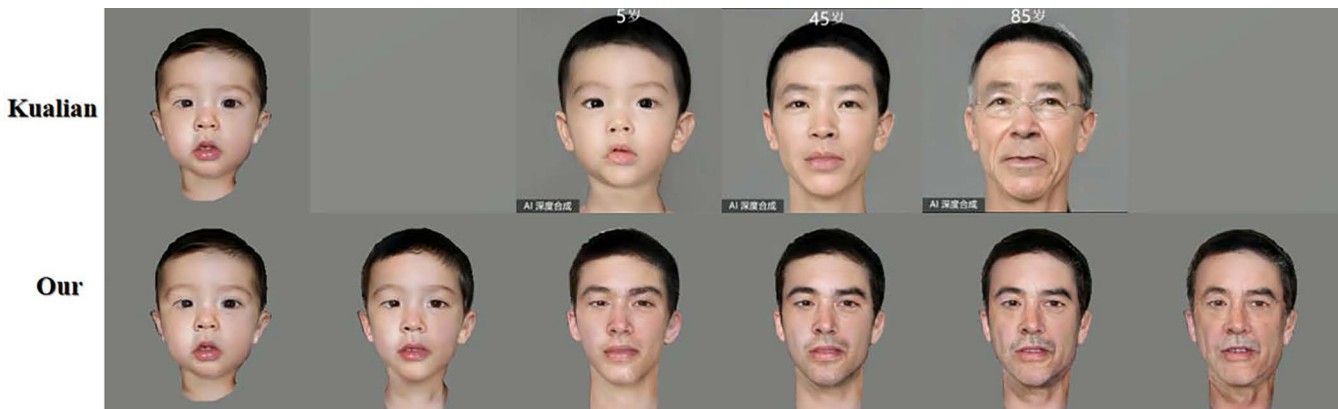

**Fig 14. Comparison with Kualian.**

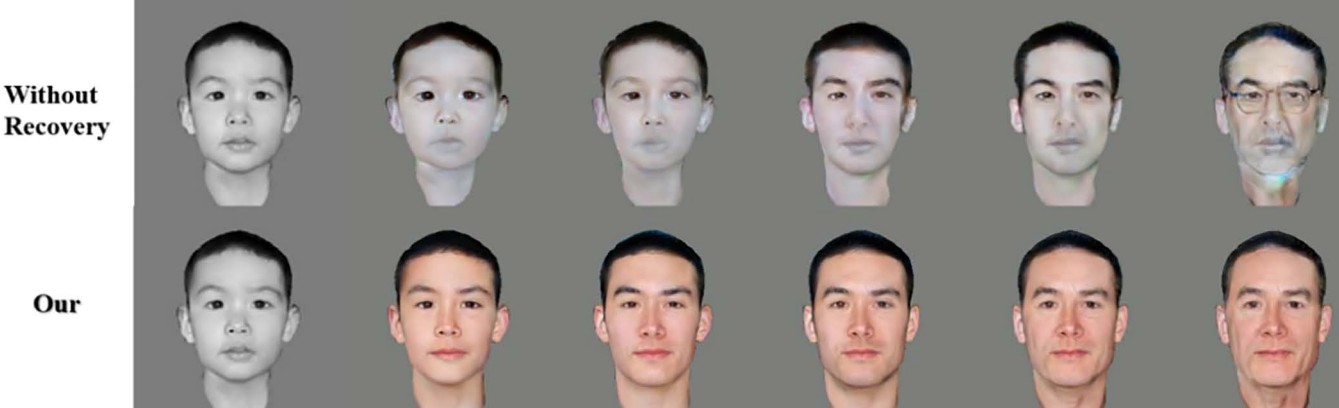

**Fig 15. Comparison with no repair.**

restoration step will lead to distorted color changes in the photo, interfering with the subsequent recognition process. After integrating the old photo restoration optimization mechanism, the face prediction effect is greatly improved.

In order to compare the impact of the old photo restoration operation on subsequent comparisons more intuitively, further analysis was performed by calculating the accuracy: the target recognition accuracy without old photo restoration optimisation was reduced to 58.9%, while the target recognition accuracy with old photo restoration optimization was 98.2%, which is a far cry from each other.

## 4.5  Self-attention model optimisation

In the context of decoupled life-cycle face synthesis, this study proposes a novel model that effectively addresses the challenge of generating realistic face images while maintaining identity consistency. The model extracts shape, texture, and identity features separately from the encoder and designs two transformation modules to simulate the nonlinear transformations of shape and texture features, respectively. This approach enables the model to reflect age sensitivity while maintaining identity consistency, as compared with the currently popular Lifelong Few-Shot (LFS) model, as shown in Fig 16.

The LFS model performs better in maintaining natural transformations of facial features but slightly lacks in the clarity and realism of facial details. In contrast, the proposed model excels in facial details, especially in simulating skin changes caused by aging. However, further improvements may be needed to ensure smooth age transitions.

The self-attention mechanism plays a crucial role in the AT-GAN model, ensuring global consistency. It guarantees that facial features (such as face shape, eye size, hair color, etc.) change gradually and consistently over time during facial age generation. Additionally, by considering long-range dependencies, the self-attention mechanism helps the model more accurately render subtle facial details, such as fine lines, wrinkles, and skin tone changes, which are important visual cues related to aging.

## 4.6  Validation and analysis of the self-attention mechanism

(1)  **Performance comparison.**  To validate the specific impact of the self-attention mechanism on model performance, a series of comparative experiments were designed. Two models were evaluated in the experiments: GAN (a baseline model without the self-attention mechanism) and AT-GAN (a model integrated with the self-attention mechanism). By

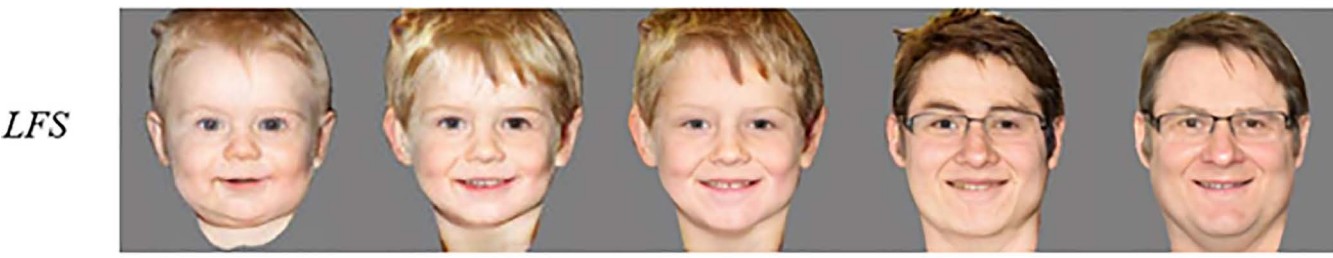
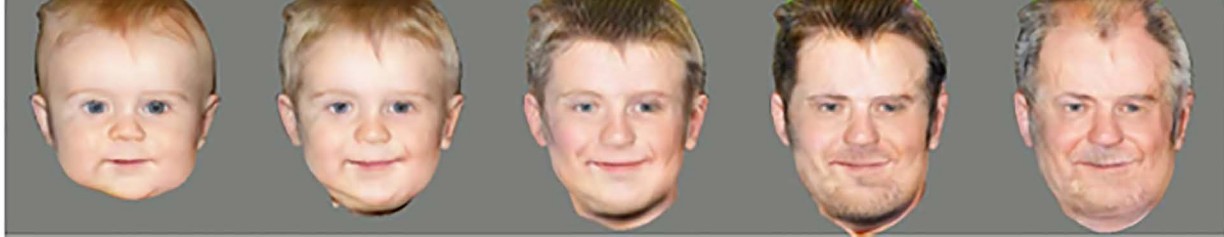

**Fig 16. Comparison of AT-GAN LFS Models.**

comparing the performance of these two models in cross-age face recognition tasks, the advantages of the self-attention mechanism were demonstrated. The specific process is as follows.

Fifty volunteers were selected, ensuring that their childhood photos were clear and of high quality. Each volunteer provided one childhood photo (as the input image) and one current-age photo (as the reference image).

A dataset containing 500 photos was constructed, including current-age photos of the 50 volunteers and 450 random face photos of non-volunteers. All photos in the dataset were preprocessed to ensure consistent resolution and format.

The model was used to predict the current-age facial images of each volunteer based on their childhood photos. The generated predicted images were matched against the 500 photos in the dataset.

In the experiment, all pre-trained parameters related to the self-attention mechanism were removed to directly compare the results. The following algorithms were compared at each anchor age category: GAN and AT-GAN. The experimental results are presented in tabular form. The table 1 shows the accuracy, recall, and F1 score on the test set, quantifying the performance differences between the methods.

The results show that the AT-GAN model outperforms the GAN model in all metrics, with an accuracy improvement of approximately 5.2%. This improvement is attributed to the self-attention mechanism, which enhances the model's ability to focus on key facial details, thereby improving the model's adaptability to age changes and the realism of the generated images.

**(2) Qualitative analysis.** To evaluate users' subjective perceptions of the results generated by different methods, a volunteer qualitative study was conducted. From the dataset of current-age photos of 50 volunteers and 450 random face photos of non-volunteers, 20 images were randomly selected, and 500 unique responses were collected from the volunteers. Users were asked to evaluate the following three aspects: (a) whether the method preserved the identity of the person in the photo; (b) how close the perceived age of the generated image was to the target age; and (c) which result was better overall.

The user study results (as shown in table 2) indicate that although GAN performs better in identity preservation, AT-GAN has a greater advantage in age accuracy and overall quality. This suggests that the self-attention mechanism significantly enhances the model's performance in cross-age face recognition tasks.

**(3) Limitations.** Unlike previous methods, our framework can simultaneously alter shape and texture to adapt to the aging process. The proposed architecture and training scheme can accurately generalize age, allowing us to generate unseen age results during training through latent space interpolation. Additionally, we introduce a new facial dataset that can be used by the visual community for various tasks.

Although our network can generalize age transformations, it has limitations in handling extreme poses, removing glasses, eliminating dense beards, and dealing with occluded faces. These issues may stem from the use of only two

**Table 1. Performance Comparison of GAN and AT-GAN in Cross-Age Face Recognition Tasks.**

| Model | Accuracy | Recall | F1 Score |
|---|---|---|---|
| GAN | 81.5% | 78.9% | 80.2% |
| AT-GAN | 86.7% | 83.4% | 85.0% |

**Table 2. User Study Results Comparison (GAN vs. AT-GAN).**

| Metric | GAN | AT-GAN |
|---|---|---|
| Identity Preservation | 18 | 12 |
| Age Difference (Average) | 22.5 | 7.2 |
| Overall Superiority | 5 | 15 |

downsampling layers in the identity encoder and the limitations of the latent identity loss. These factors restrict the network's generalization ability in these scenarios.

## Conclusion

This manuscript introduces a novel cross-age face recognition methodology utilizing a generative adversarial network (GAN), designed to extract pivotal features from an individual's early photographs and predict facial transformations, thereby facilitating the construction of a current facial portrait. The proposed AT-GAN network architecture integrates a self-attention mechanism to bolster the model's proficiency in identifying and preserving essential facial characteristics, while also refining old photo restoration techniques to augment the precision and robustness of facial predictions. Despite the promising outcomes of the current research, certain limitations and prospective research avenues remain. Primarily, this study intends to further refine the AT-GAN model to enhance its generalizability and real-time processing capabilities in more intricate scenarios. The accuracy is somewhat elevated due to the constraints imposed by the current facial dataset's scale. Subsequently, this research will investigate the integration of hard and soft biometric information more effectively to further elevate the model's recognition accuracy. Moreover, the study will also delve into addressing recognition challenges posed by occlusions, variations in lighting, and differences in facial expressions to strengthen the model's robustness in practical applications.

## Author contributions

**Conceptualization:** Xingyuan Peng, Ruoyi Xu.

**Formal analysis:** Guangxuan Chen.

**Methodology:** Xingyuan Peng.

**Software:** Ruoyi Xu.

**Writing – original draft:** Guangxuan Chen, Xingyuan Peng.

**Writing – review & editing:** Guangxuan Chen.

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
