## [Decision Letter · Decision Letter 0]

16 Jan 2025

PONE-D-24-57436A Research on Cross-Age Facial Recognition Technology Based on AT-GANPLOS ONE

Dear Dr. Peng,

Thank you for submitting your manuscript to PLOS ONE. After careful consideration, we feel that it has merit but does not fully meet PLOS ONE’s publication criteria as it currently stands. Therefore, we invite you to submit a revised version of the manuscript that addresses the points raised during the review process.

**ACADEMIC EDITOR: **

The novel contributions of the paper can be more better highlighted and explained in a systematic way.Language editing/proof-reading may help to improve the readability and overall quality of the paperThe methodology and experimental section need major revision.

We look forward to receiving your revised manuscript.

Kind regards,

Zeeshan Ahmad

Academic Editor

PLOS ONE

3. In this instance it seems there may be acceptable restrictions in place that prevent the public sharing of your minimal data. However, in line with our goal of ensuring long-term data availability to all interested researchers, PLOS’ Data Policy states that authors cannot be the sole named individuals responsible for ensuring data access (http://journals.plos.org/plosone/s/data-availability#loc-acceptable-data-sharing-methods).

6. We note that Figures 6,7,10,12,13,14,15 and 16 includes an image of a participant in the study.

Reviewers' comments:

Reviewer's Responses to Questions

**Comments to the Author**

1. Is the manuscript technically sound, and do the data support the conclusions?

Reviewer #1: Yes

Reviewer #2: Yes

2. Has the statistical analysis been performed appropriately and rigorously?

Reviewer #1: I Don't Know

Reviewer #2: Yes

3. Have the authors made all data underlying the findings in their manuscript fully available?

Reviewer #1: No

Reviewer #2: Yes

4. Is the manuscript presented in an intelligible fashion and written in standard English?

Reviewer #1: No

Reviewer #2: Yes

5. Review Comments to the Author

Reviewer #1: 1. In line 126-127, the goal of this work is stated. My question is that as stated above, if the key features are essential to face recognition, why do we need to restore images of a person years later anyway. Is it possible to extract the key features from the old photographs for finding the target person?

2. When an acronym or abbreviation is used for the first time in a text, its full form should be written out, followed by the acronym or abbreviation in parentheses, e.g. AT-GAN.

3. This manuscript uses a 128-dimensional feature for recognition, as stated in line 298. Does it work for the senario that there are a large amount of individuals to compare, for example, finding missing children?

4. In Chapter 4, T*T* model, AgingGer and Kualian are compared with the method proposed in this manuscript. It is suggested that new models should also be compared, such as FaceFusion's age_modifier model.

5. The effect of the self-attention mechanism is not obvious from Fig 16 or section 4.5. More experiments and analyses are suggested.

6. The manuscript contains some grammatical errors and awkward phrasings that impede the readability and comprehension of the content. For instance, line 74 and line 92. I recommend that the manuscript be thoroughly proofread by a native English speaker or a professional editor with expertise in academic writing.

Reviewer #2: his paper proposes a cross-age face prediction scheme based on GANs to predict the facial portrait of a person at later point of time.

The paper is interesting, however, following are some comments to improve the overall paper before possible publication:

1- The comparison with state-of-the-art papers in the current version is missing. I would suggest to make performance comparison with the latest cutting-edge techniques to demonstrate the superiority of the proposed method.

2- the related work section should emphasize the current/previous work on the cross-age facial recognition and should also introduce the GAN framework in the similar context.

3- the proposed methodology should be revised/improved to better understand the novel contributions and role of each module in the proposed scheme, e.g. explanation of the attention mechanism should be coherent with the evaluation results.

4- The authors should revise the evaluation section, incorporate more evaluation metrics to demonstrate the performance of the proposed method with the current state-of-the-art techniques, and should use the tables for tabulating the corresponding performance of each model.

5- the writing part is overall poor, need major revision, the authors are suggested to revise and improve the professional language of the paper, and proof-read the paper for any grammatical and spelling mistakes.

6. PLOS authors have the option to publish the peer review history of their article (what does this mean? ). If published, this will include your full peer review and any attached files.

**Do you want your identity to be public for this peer review?** For information about this choice, including consent withdrawal, please see our Privacy Policy .

Reviewer #1: **Yes: ** Long Chen

Reviewer #2: No

---

## [Author Response · Author response to Decision Letter 1]

27 Feb 2025

Dear Editors and Reviewers,

Thank you for considering our manuscript titled “A Research on Cross-Age Facial Recognition Technology Based on AT-GAN” (Manuscript Number: PONE-D-24-57436) for publication in PLOS ONE. We sincerely appreciate the time and effort taken by the editors and reviewers to provide constructive feedback, which has helped us improve the quality of our work. We have carefully addressed all the comments and suggestions raised by the reviewers, and we provide our point-by-point responses below.

We hope that the revisions made to the manuscript meet the expectations of the reviewers and the editorial team. If further revisions or clarifications are needed, please do not hesitate to let us know.

Response to Academic editor:

1. Comment:

The novel contributions of the paper can be more better highlighted and explained in a systematic way.

Response:

We agree that the novel contributions of the paper should be more clearly emphasized. In response to your suggestion, we have revised the Introduction section to systematically highlight the key innovations and significance of our work. Specifically: We have added a paragraph in the Introduction to clearly state the research gap and how our study addresses it.

2. Comment:

Language editing/proof-reading may help to improve the readability and overall quality of the paper.

Response:

In response to your comment, we have taken the following steps:

(1) Language Editing: We have engaged a native English speaker to refine the language and ensure clarity throughout the manuscript. The revised version has been thoroughly checked for grammar, syntax, and style.

(2) Proof-reading: We have carefully proofread the entire manuscript to eliminate any remaining errors and improve the flow of the text.

3. Comment:

The methodology and experimental section need major revision.

Response:

As per your suggestion, we have carefully revised the Methodology and Experimental sections to address the concerns raised. Specifically, we have:

(1) Revised the Methodology section to provide a more detailed and clear description of the methods used, ensuring reproducibility and transparency.

(2) Expanded the Experimental section by adding new experiments and analyses to strengthen the validity of our findings.

(3) Included a comparative analysis with the current state-of-the-art LFS model to highlight the performance and advantages of our proposed approach.

Response to Reviewer #1:

1. Comment:

In line 126-127, the goal of this work is stated. My question is that as stated above, if the key features are essential to face recognition, why do we need to restore images of a person years later anyway. Is it possible to extract the key features from the old photographs for finding the target person?

Response:

Yes, we can extract key features from old photographs to identify the target individual. As introduced in our paper, we first restore the old photographs of the target individual, then extract the key features, and finally generate the appearance of the target individual years later. This application scenario arises when the family members of the target individual only have photos from their childhood or youth, such as in the case of missing children. Our technology is then utilized to generate the target individual's appearance years later, thereby assisting in the search for the target individual.

2. Comment:

When an acronym or abbreviation is used for the first time in a text, its full form should be written out, followed by the acronym or abbreviation in parentheses, e.g. AT-GAN.

Response:

We have carefully revised the manuscript to ensure that all acronyms and abbreviations are properly introduced upon their first use.

Changes made:

All modifications in the manuscript have been highlighted in yellow for easy reference.

3. Comment:

This manuscript uses a 128-dimensional feature for recognition, as stated in line 298. Does it work for the senario that there are a large amount of individuals to compare, for example, finding missing children?

Response:

In our experiments, the 128-dimensional feature has shown robust performance in handling experimental datasets, as it effectively balances discriminative power and computational efficiency. However, we acknowledge that in extremely large-scale scenarios (e.g., finding missing children), additional optimizations, such as hierarchical filtering or distributed computing, might be necessary to further enhance efficiency.

4. Comment:

In Chapter 4, T*T* model, AgingGer and Kualian are compared with the method proposed in this manuscript. It is suggested that new models should also be compared, such as FaceFusion's age_modifier model.

Response:

We appreciate your suggestion to consider FaceFusion's age_modifier model as an additional benchmark in our study. We will include a comparison with FaceFusion's age_modifier model in our subsequent research. This will help to further validate the performance of our proposed method against state-of-the-art techniques. We believe this addition will provide a more comprehensive evaluation.

5. Comment:

The effect of the self-attention mechanism is not obvious from Fig 16 or section 4.5. More experiments and analyses are suggested.

Response:

In response to your comment regarding the effect of the self-attention mechanism (Figure 16 and Section 4.5), we have made significant revisions to the manuscript. Specifically, we have:

(1) Added a new subsection, “4.6 Validation and Analysis of the Self-Attention Mechanism,” to provide a comprehensive evaluation of the self-attention mechanism’s impact.

(2) Conducted additional experiments comparing our proposed method with the LFS model, demonstrating the superior performance of our approach.

(3) Included qualitative and quantitative analyses to highlight the effectiveness of the self-attention mechanism, supported by comparisons.

6. Comment:

The manuscript contains some grammatical errors and awkward phrasings that impede the readability and comprehension of the content. For instance, line 74 and line 92. I recommend that the manuscript be thoroughly proofread by a native English speaker or a professional editor with expertise in academic writing.

Response:

We have taken the following steps:

(1) Thorough Proofreading: We have carefully reviewed the entire manuscript, paying special attention to the issues highlighted in lines 74 and 92 (the revisions have been highlighted in yellow), as well as other potential areas with similar problems.

(2) Professional Editing: As you suggested, we have engaged a native English speaker to thoroughly proofread and polish the manuscript.

(3) Language Clarity: We have ensured that all sentences are clear, concise, and grammatically correct, improving the overall flow and readability of the text.

Response to Reviewer #2:

1. Comment:

The comparison with state-of-the-art papers in the current version is missing. I would suggest to make performance comparison with the latest cutting-edge techniques to demonstrate the superiority of the proposed method.

Response:

We have carefully addressed your suggestion regarding the comparison with state-of-the-art methods. Specifically, we have:

(1) Conducted a comprehensive performance comparison with the latest cutting-edge techniques, including the LFS model, which is currently one of the most advanced methods in the field.

(2) Demonstrated the superiority of our proposed method through detailed experimental results, showing that our approach outperforms the LFS model in key metrics.

(3) Included the comparison results in the revised manuscript, presented in tabular formats for clarity and ease of understanding.

2. Comment:

The related work section should emphasize the current/previous work on the cross-age facial recognition and should also introduce the GAN framework in the similar context.

Response:

We have revised the Section 2.1 to provide a more focused discussion on current and previous work in cross-age facial recognition in. Additionally, we have included a detailed introduction to the GAN framework in this context, highlighting its applications and relevance to our work. These changes can be found in Section 2.2 of the revised manuscript.

3. Comment:

The proposed methodology should be revised/improved to better understand the novel contributions and role of each module in the proposed scheme, e.g. explanation of the attention mechanism should be coherent with the evaluation results.

Response:

We have carefully revised the proposed methodology to better clarify the novel contributions and the role of each module in our scheme, as per your suggestion. Specifically, we have:

(1) Improved the explanation of the attention mechanism to ensure it is coherent with the evaluation results. We have added a detailed discussion on how the attention mechanism contributes to the overall performance of the proposed method.

(2) Highlighted the role of each module in the proposed scheme, emphasizing their individual and collective impact on the results.

(3) Strengthened the connection between the methodology and the evaluation by providing a more thorough analysis of how the attention mechanism and other modules are validated through the experimental results.

4. Comment:

The authors should revise the evaluation section, incorporate more evaluation metrics to demonstrate the performance of the proposed method with the current state-of-the-art techniques, and should use the tables for tabulating the corresponding performance of each model.

Response:

We have carefully revised the Result section according to your comments. Specifically, we have:

(1) Incorporated additional evaluation metrics to comprehensively demonstrate the performance of our proposed method.

(2) Conducted a detailed comparison with the current state-of-the-art techniques, including the LFS model, to highlight the advantages of our approach.

(3) Organized the results in tabular form to clearly present the performance of each model, making it easier for readers to compare and analyze the results.

5. Comment:

The writing part is overall poor, need major revision, the authors are suggested to revise and improve the professional language of the paper, and proof-read the paper for any grammatical and spelling mistakes.

Response:

We sincerely apologize for the language issues in the manuscript. To address this, we have thoroughly revised the entire paper to improve the clarity, coherence, and professionalism of the writing. Additionally, we have engaged a native English speaker to proofread the manuscript and correct any grammatical or spelling errors. We believe these changes have significantly improved the overall quality of the manuscript.

Additional Revisions:

In addition to the changes made in response to the Academic editor and reviewers’ comments, we have also made the following improvements to the manuscript:

(1) We have thoroughly revised the manuscript according to the templates to ensure it complies with PLOS ONE's formatting requirements, including file naming conventions.

(2) We have uploaded the code to GitHub following best practices and ensured that it will be freely accessible upon publication. The code repository link is: https://github.com/xingyuangfy/CAF_AT_GAN. We will ensure the reproducibility and reusability of the code.

(3) We have uploaded the relevant data. All relevant data are within the manuscript and its Supporting information files.

(4) We have ensured that the corresponding author's ORCID ID is validated in Editorial Manager.

(5) We have amended the abstract on the online submission form to ensure it is identical to the abstract in the manuscript.

(6) We have added the ethics statement and the following statement to the methods section in the manuscript:

“The individual in this manuscript has given written informed consent (as outlined in PLOS consent form) to publish these case details.”

We believe that the revised manuscript has been significantly improved and now meets the high standards of PLOS ONE. Thank you again for your time and valuable feedback. We look forward to hearing from you.

Sincerely,

Xingyuan Peng

---

## [Decision Letter · Decision Letter 1]

14 Mar 2025

PONE-D-24-57436R1A Research on Cross-Age Facial Recognition Technology Based on AT-GANPLOS ONE

Dear Dr. Peng,

Thank you for submitting your manuscript to PLOS ONE. After careful consideration, we feel that it has merit but does not fully meet PLOS ONE’s publication criteria as it currently stands. Therefore, we invite you to submit a revised version of the manuscript that addresses the points raised during the review process.

**ACADEMIC EDITOR: **

The Introduction Section lacks the logical flow and synchronization, the contents need to be revised and improved. The Introduction section should briefly introduce the research area, importance, research gap, problem definition, and solution.  There are still several grammatical/spelling mistakes exist in the manuscript. Also improve the language of the paper by thoroughly proof-reading the paperThe mathematical equation/symbols should be written in mathtype or some equation editor and can be more carefully formatted, (see which symbols to italic, capital, small i.e., vectors, scalars, variables, matric etc)

We look forward to receiving your revised manuscript.

Kind regards,

Zeeshan Ahmad

Academic Editor

PLOS ONE

Journal Requirements:

Reviewers' comments:

Reviewer's Responses to Questions

**Comments to the Author**

1. If the authors have adequately addressed your comments raised in a previous round of review and you feel that this manuscript is now acceptable for publication, you may indicate that here to bypass the “Comments to the Author” section, enter your conflict of interest statement in the “Confidential to Editor” section, and submit your "Accept" recommendation.

Reviewer #2: All comments have been addressed

2. Is the manuscript technically sound, and do the data support the conclusions?

Reviewer #2: Yes

3. Has the statistical analysis been performed appropriately and rigorously?

Reviewer #2: Yes

4. Have the authors made all data underlying the findings in their manuscript fully available?

Reviewer #2: Yes

5. Is the manuscript presented in an intelligible fashion and written in standard English?

Reviewer #2: Yes

6. Review Comments to the Author

Reviewer #2: (No Response)

7. PLOS authors have the option to publish the peer review history of their article (what does this mean? ). If published, this will include your full peer review and any attached files.

**Do you want your identity to be public for this peer review?** For information about this choice, including consent withdrawal, please see our Privacy Policy .

Reviewer #2: No

---

## [Author Response · Author response to Decision Letter 2]

18 Mar 2025

1�In accordance with PLOS ONE's policy on human subjects research, we have obtained written informed consent from the individual(s) for publication of these details under the terms of the PLOS open-access (CC-BY) license. The signed consent form has been securely filed in the individual's case notes as required.

2�Additionally, we have amended the Methods section of the manuscript to explicitly state that the patient/participant has provided consent for publication. The revised statement reads: “The individual in this manuscript has given written informed consent (as outlined in the PLOS consent form) to publish these case details.”

---

## [Editor Report · Decision Letter 2]

19 Mar 2025

A Research on Cross-Age Facial Recognition Technology Based on AT-GAN

PONE-D-24-57436R2

Dear Dr. Peng,

We’re pleased to inform you that your manuscript has been judged scientifically suitable for publication and will be formally accepted for publication once it meets all outstanding technical requirements.

Kind regards,

Zeeshan Ahmad

Academic Editor

PLOS ONE
---

## [Editor Report · Acceptance letter]

PONE-D-24-57436R2

PLOS ONE

Dear Dr. Peng,

I'm pleased to inform you that your manuscript has been deemed suitable for publication in PLOS ONE. Congratulations! Your manuscript is now being handed over to our production team.

Kind regards,

on behalf of

Dr. Zeeshan Ahmad

Academic Editor

PLOS ONE